# The Anti-*Candida* Activity of *Tephrosia apollinea* Is More Superiorly Attributed to a Novel Steroidal Compound with Selective Targeting

**DOI:** 10.3390/plants11162120

**Published:** 2022-08-15

**Authors:** Naglaa S. Ashmawy, Eman M. El-labbad, Alshaimaa M. Hamoda, Ali A. El-Keblawy, Abdel-Nasser A. El-Shorbagi, Kareem A. Mosa, Sameh S. M. Soliman

**Affiliations:** 1Sharjah Institute for Medical Research, University of Sharjah, Sharjah P.O. Box 27272, United Arab Emirates; 2Faculty of Pharmacy, Ain Shams University, El-Abaseya, Cairo 11566, Egypt; 3Department of Pharmaceutical Sciences, College of Pharmacy, Gulf Medical University, Ajman P.O. Box 4184, United Arab Emirates; 4College of Medicine, University of Sharjah, Sharjah P.O. Box 27272, United Arab Emirates; 5Faculty of Pharmacy, Assiut University, Assiut 71526, Egypt; 6Department of Applied Biology, University of Sharjah, Sharjah P.O. Box 27272, United Arab Emirates; 7Department of Medicinal Chemistry, College of Pharmacy, University of Sharjah, Sharjah P.O. Box 27272, United Arab Emirates; 8Department of Biotechnology, Faculty of Agriculture, Al-Azhar University, Cairo 11651, Egypt

**Keywords:** *Tephrosia apollinea*, anti-*Candida*, hot arid desert, novel steroid, natural optimization

## Abstract

*Tephrosia* is widely distributed throughout tropical, subtropical, and arid regions. This genus is known for several biological activities, including its anti-*Candida* activity, which is mainly attributed to prenylated flavonoids. The biological activities of most *Tephrosia* species have been studied, except *T. apollinea*. This study was conducted to investigate the underlying anti-*Candida* activity of *T. apollinea*, wildly grown in the United Arab Emirates (UAE). The *T. apollinea* plant was collected, dried, and the leaves were separated. The leaves were ground and extracted. The dried extract was subjected to successive chromatography to identify unique phytochemicals with a special pharmacological activity. The activity of the compound was validated by homology modeling and molecular docking studies. A novel steroidal compound (ergosta-6, 8(14), 22, 24(28)-tetraen-3-one) was isolated and named TNS. In silico target identification of TNS revealed a high structural similarity with the *Candida* 14-α-demethylase enzyme substrate. The compound exhibited a significant anti-*Candida* activity, specifically against the multi-drug-resistant *Candida auris* at MIC_50_, 16 times less than the previously reported prenylated flavonoids and 5 times less than the methanol extract of the plant. These findings were supported by homology modeling and molecular docking studies. TNS may represent a new class of *Candida* 14-α-demethylase inhibitors.

## 1. Introduction

*Tephrosia apollinea* (Delile) DC., a perennial evergreen shrub, is one of more than 350 species in the genus *Tephrosia*. The genus belongs to the family Fabaceae. The plants of this genus are distributed in tropical, subtropical, and arid regions [1]. *T. apollinea* is widely distributed in southwest Asia, northwestern India, and northeast Africa [2]. The plant is native to the UAE and is widely grown in the lower mountains [3]. Based on taxonomical studies, *Tephrosia* was classified into three subgenera, including Marconyx such as *T. tenuis*, Brissonia such as *T. candida*, and Reineria, which includes the remaining *Tephrosia* species [4].

Phytochemical investigations revealed the presence of several phytoconstituents in *Tephrosia* species. Flavonoids are the most commonly isolated and identified compounds in the genus, in addition to other classes of compounds, including rotenoids, terpenoids, sterols, essential oils, and fixed oils [5]. Few studies have evaluated the presence of volatile oil and fixed oil in this genus. Sesquiterpenes, diterpenoids, furano chalchones, and fatty acids have been identified in *T. purpurea* [6].

The pharmacological activities of different *Tephrosia* spp. have been extensively studied. The biological activities associated with the genus are antioxidant [7], anti-diabetic [7], anti-ulcer [8], wound healing [9], anti-inflammatory [10], anticancer [11], insecticidal [12], anti-protozoal [13], and anti-fungal [5]. The anti-fungal activity of the plant species was mostly reported against *Cladosporium cucumerinum* [14], *Aspergillus niger,* and *Candida albicans* [15].

Plants from the genus *Tephrosia* have been used for major traditional medicinal uses. *T. purpurea* is a well-known species among the genus *Tephrosia* for its importance in folk and traditional therapy; the whole plant is widely used for the treatment of leprosy, ulcers, and inflammatory conditions, including asthma and chronic bronchitis [16]. The dried herb is widely used as a tonic, laxative, and diuretic [17]. Plant extracts are used to alleviate both liver and spleen disorders [6]. In fact, traditional healers use various parts of this plant to cure all types of wounds [18].

*T. purpurea*, *T. toxicaria*, *T. candida*, *T. elata*, and *T. villosa* are among the species whose pharmacological and biological activities have been studied [19]. However, few studies have been conducted on *T. apollinea*. The antifungal activity of *T. apollinea* was attributed to prenylated flavonoids, including, in particular, Tephroapollin-F and Lanceolatin-A, but at a higher MIC_50_ value of 4 mg/mL [20]. The antifungal activity of the ethanolic root extract of a population of *T. apollinea* growing in Egypt was tested using the cup-plate screening method against three fungi, *Aspergillus flavus*, *A. fumigatus,* and *Penicillium chrysogenum*. The extract showed a high inhibition activity against the three tested organisms with a range of 19–22 mm diameter for the inhibition zone [21]. 

*T. apollinea* growing in the arid desert of the UAE faces very harsh conditions, such as drought and high temperatures that reach up to 45 ℃ in summer. Therefore, we hypothesized that such a climate may develop specific phytochemicals for adaption to such conditions. In a similar study, we identified a novel anticancer compound from *Ziziphus spina-christi* growing in the hot arid desert of the UAE [22]. Herein, we correlate the traditional antifungal activity of *T. apollinea*, growing in the hot arid environment of the UAE, to a novel phytochemical using chemistry and biology techniques. 

## 2. Results

### 2.1. Discovery of Novel Phytochemicals from T. apollinea

*T. apollinea* leaf methanol extract was subjected to successive chromatography to obtain four compounds (**1**–**4**), as described in the experimental section. The structures of the isolated compounds were identified by spectroscopy (Appendix A) and through comparison with previously reported data (Figure 1 and Table 1). The compounds were identified as ergosta-6, 8 (14), 22-tetraen-3one (compound **1**, 100 mg), lanceolatin A (compound **2**, 180 mg), and a mixture of stigmasterol and *β*-sitosterol (compound **3** and **4**, 200 mg) (Figure 1 and Table 1). On the other hand, compound# **1**, ergosta-6, 8 (14), 22-tetraen-3one, was the first time to be reported in nature. Compound **1** was given the name TNS.

TNS was purified as white crystals. The molecular formula was determined as C_28_H_40_O by ESI-MS^+^. ^1^HNMR spectroscopy exhibited a downfield region consisting of six olefinic H-atoms at *δ_H_* 5.78 (*d*, *J* = 15.7), 5.13 (*dd*, *J* = 15.8, 9.9), 5.11 (*dd*, *J* = 15.7, 8) 5.25 (*d*, *J* = 9.9), and 4.79 (*d*, *J* = 2.3), 4.74 (*d*, *J* = 2.3), while the upfield region of the spectrum showed five methyl group signals at *δ_H_* 0.89 (*d*, *J* = 6.8), 0.86 (*s*), 0.81 (*d*, *J* = 6.8, 6-H), and 0.86 (*s*), together with many CH_2_ and CH signals in the range of 2.28–1.40 (Table 1). ^13^CNMR spectroscopy showed 28 C-atom signals, including one carbonyl signal at *δ*c 207.2 and olefinic C signals in the range of *δ*c 149.1–109.1 (Table 1). These spectroscopic data suggest that TNS is a steroidal compound. The structure of TNS was confirmed by two-dimensional NMR spectral analysis. The HMBC spectrum exhibited a cross peak correlating H-2 (*δ*_H_ 2.11) with C-3 (*δ*_C_ 207.2), confirming the absence of a double bond at position 4. A detailed two-dimensional NMR analysis, including HSQC, COSY, and HMBC, was performed and it confirmed the structure of TNS as ergosta-6, 8(14), 22, 24(28)-tetraen-3-one (Figure 2 and Table 1).

### 2.2. TNS Compound Showed Promising Anti-Candida Activity

#### 2.2.1. In Silico Target Analysis of TNS Compound

Screening the chemical features of the newly discovered compound revealed a high structural similarity to the *Candida* 14-α-demethylase enzyme substrates, such as lanosterol, eburicol, and obtusifoliol. The common 3D pharmacophore features between lanosterol, eburicol, and obtusifoliol were constructed using Biovia discovery studio^®^ common feature pharmacophore protocol (Figure 3A). The generated pharmacophore model was composed of six features, including five hydrophobic features and one hydrogen bond acceptor. Mapping the lanosterol, eburicol, obtusifoliol, and TNS structures with the generated pharmacophore showed a similar alignment pattern with the pharmacophore at a fitting value ranging from 5.99 to 4.89 out of 6 (Figure 3B–E). Lanosterol, eburicol, and obtusifoliol structures contained an OH group at position 3 as a hydrogen bond acceptor feature, steroid ring B as hydrophobic point 1, C4-methyl as hydrophobic point 2, C-21 methyl as hydrophobic point 3, C 24 and C28 C = C as hydrophobic point 4, and the isopropyl group at C25–C27 as hydrophobic point 5. The newly isolated TNS compound showed almost a pharmacophoric feature and alignment pattern similar to lanosterol, eburicol, and obtusifoliol, except for a missing alignment with the hydrophobic point 2 due to the absence of alkyl substitution at C-4, leading to a decrease in fitting value to 4.69 out of 6 (Figure 3E). Based on the clear 3D pharmacophoric feature similarity between TNS and the 14-α-demethylase enzyme substrates and the previously reported anti-*Candida* activity of the extracts from the genus *Tephrosia* [19,23], TNS was expected to exhibit an antifungal activity.

#### 2.2.2. Anti-*Candida* Activity of TNS Compound

The anti-*Candida* activity of TNS was tested against *C. auris* and *C. albicans*. The results indicated that the compound caused significant inhibition to the growth of multi-drug-resistant *C. auris* (Figure 4A) and *C. albicans* (Figure 4B) at MIC_50_ 200 and 700 µg∕mL, respectively. TNS showed significant promising activities on *C. auris* when compared to fluconazol, which was employed as a positive control (two-way ANOVA, *p* < 0.001) (Figure 4A), while a reduced activity on *C. albicans* (Figure 4B). The results indicate that TNS did not only show a significant antifungal activity, but also showed a selective promising activity against *C. auris*. TNS at its MIC_50_ showed no toxicity to the normal human dermal fibroblast cell line (HDF) (Figure 4C). On the other hand, the plant methanolic extract showed a minimal (15–20%) inhibition activity to both *C. auris* and *C. albicans* at ~5 times the MIC_50_ of TNS. Furthermore, this concentration caused more than 70% killing to mammalian cells. Collectively, these results indicate that TNS isolated from *T. apollinea* exhibited a significant anti-*Candida* activity, particularly against *C. auris*.

#### 2.2.3. Binding Simulation of TNS with *C. Auris* 14-α-Demethylase Validate the Anti-*Candida* Activity of TNS

As the anti-*Candida* activity of TNS may be attributed to possible binding inhibition with the 14-α-demethylase enzyme, as suggested by our pharmacophore modeling, a molecular docking study of TNS against *C. auris* 14-α-demethylase was conducted using Discovery Studio 2.5. Because the crystal structure of *C. auris* 14-α-demethylase was not resolved until now, a homology model was conducted to simulate the 3D structure. The data homology model was reported as a reliable tool to create a 3D structure from a given amino acid sequence [24]. The X-ray crystal structure of *S. cerevisiae* 14-α demethylase bound with lanosterol (PDB: 4LXJ) was used in this study as a template [25]. The sequence alignment between *C. auris* 14-α-demethylase (NCBI Reference Sequence: XP_028891800.1) and that of *S. cerevisiae* using align sequence protocol in Discovery Studio 2.5 showed sequence similarity and identity of 83.4 % and 64.9 %, respectively (Figure 5). This indicates a higher reliability of the model, as >30% of the amino acid identity between the 3D model and template was identified [26].

The 3D structure of *C. auris* 14-α-demethylase enzyme was obtained using the Build Homology Model Protocol in Discovery Studio 2.5 using the co-crystallized heme and lanosterol (Figure 6). A 3D model of the least PDF Total Energy and DOPE Score were included in our study. Further validation of the modeled protein was conducted using MODELER [27] and PROFILE 3D protocols in D.S 2.5. (Figure 7). A Ramachandran plot was used to assess the energetically allowed regions of the model (Figure 8). Only 1.9% of the residues were in the disallowed region; these amino acids were located in the lope regions away from the binding site. The model protein showed good alignment with the template 14-α demethylase bound with lanosterol (PDB: 4LXJ) with a total protein sequence alignment RMSD equal 0.942 (Figure 9).

To simulate the binding pattern of TNS with the *C. auris* 14-α-demethylase enzyme, Discovery Studio 2.5 CDOCKER protocol was employed (Figure 10). The binding energy calculation of the TNS−*C. auris* 14-α-demethylase enzyme complex and lanosterol−*C. auris* 14-α-demethylase enzyme complex were conducted using calculate ligand-binding protocol in Discovery Studio 2.5 (Table 2). TNS showed a comparable binding energy to the natural substrate lanosterol via multiple hydrophobic interactions in the substrate binding site. Furthermore, TNS lacks 14-α methyl group, which is the key reaction site of 14-α-demethylase enzyme, suggesting TNS binding to 14-α-demethylase does not require demethylation (Figure 11).

## 3. Discussion

In this study, we isolated four compounds from *T. apollinea*, growing in the UAE, three of which were previously reported from the genus *Tephrosia,* but not from *T. apollinea*. These are lanceolatin A [21], stigmasterol, and *β*-sitosterol [28]. Furthermore, compound# **1** (TNS), ergosta-6, 8 (14), 22-tetraen-3one, has never been reported in nature. TNS is structurally very related to a previously identified compound (22E)-ergosta-4,6,8(14),22,24(28)-pentaen-3-one, isolated from the fermentation broth of a gorgonian-derived *Aspergillus* sp. fungus [29]. Comparing the NMR data of TNS with the previously reported compound indicated that both compounds have the same tetracyclic steroidal nucleus. The main difference between the two compounds is the absence of a double bond at position 4 in TNS, which is confirmed by the absence of the olefinic H-atom signal at position 4 (*δ*_H_ 5.7 (*s*)) and its corresponding olefinic C-atom (*δ*_C_ 123.2), and the absence of an olefinic C-atom at position 5 (*δ*_C_ 164.6). The absence of this double bond leads to an upfield shift of the carbonyl group at position 3 in TNS to appear at δc 207.2 instead of 199.8 in the reported compound.

The similarity in structures between TNS with that reported from the *Aspergillus* sp. might represent an evolutionary link between two different kingdoms, Plantae and Fungi. On the other hand, the double bond difference in the structure of both compounds may be attributed to the climate difference [22,30]. TNS is isolated from *T. apollinea* growing in the hot summer of the arid subtropical desert of UAE, where the maximum temperature reaches up to ~50 °C for several days (the average maximum temperature is 39.1 °C). However, the other related compound is isolated from a gorgonian-derived fungus. Gorgonia is a genus of soft corals, sea fans growing in the tropics and subtropics regions of the world [31]. Another explanation may suggest the contribution of an endophytic fungus in the production of TNS from *T. apollinea*. Various endophytic fungi were identified in different plant species for the purpose of a complementary function by promoting the production of unique bioactive compounds [22,32]. Interestingly, Halo et al. (2018) reported the isolation of *Aspergillus* endophyte from *T. apollinea* collected from Sultanate of Oman, a neighboring country to the UAE that possesses the same climate condition [33]. Ergostane-type steroids have been identified in both fungi and plants. Fungal ergosteroids have been widely isolated and reported [34,35], including those isolated from the Korean wild mushroom *Xerula furfuracea* [36], those isolated from *Lasiodiplodia pseudotheobromae* fungus [37], and those isolated from *Ganoderma lingzhi* [38]. On the other hand, ergostane-type steroids have been also isolated from plants such as *Cratylia mollis* leaves (Fabaceae) [39], *Entandrophragma angolense* (Meliaceae) [40], and *Physalis peruviana* (Solanaceae) [41]. Further studies are required to confirm or exclude the aforementioned hypotheses. The presence of TNS should also be assessed in other *Tephrosia* species growing in the UAE, including *T. haussknechtii* Bornm., *T. nubica* (Boiss.) Baker, *T. persica* Boiss., and *T. uniflora* Pers.

The pharmacophoric features of TNS confirmed its high structural similarity to the *Candida* 14-α-demethylase enzyme substrates including lanosterol, eburicol, and obtusifoliol. The 14-α-Demethylase enzyme (CYP51, Erg11, or LDM) is a well-established target for anti-fungal drug discovery, particularly azole drugs [42,43]. Demethylation of lanosterol at 14-α position is a rate-limiting step in the biosynthesis of fungal ergosterol [44]. The inhibition of 14-α-demethylase enzyme (Erg11p) can lead to ergosterol depletion from the fungal membrane, thus altering the fluidity of the fungal lipid bilayer. This is associated with the accumulation of toxic metabolites that have been reported as fungistatic compounds [45] to many pathogenic fungi such as *Candida* spp. [46] and *Aspergillus* spp. [47]. Therefore, the anti-*Candida* activity of TNS was expected.

The anti-*Candida* activity of extracts from the genus *Tephrosia* has been reported previously by different researchers [19,23]. Although the antifungal activity of *T. apollinea* was attributed to prenylated flavonoids at MIC_50_ 4mg/mL [20], here, we reported that TNS showed a more potent anti-*Candida* activity at MIC_50_ of ~6 and 20 times less against *C. albicans* and *C. auris*, thus indicating the potency and selectivity of the compound. TNS also showed ~4 and 23 times less activity against *C. albicans* and *C. auris* when compared with the plant methanol extract, while the plant itself contained ~0.2 mg/gm dry weight. These results deserve more future investigation to identify the selectivity of the compound. Furthermore, TNS at its MIC_50_ showed no toxicity to mammalian cells compared with the 70% killing effect due to the methanolic extract, indicating the wide safety profile of TNS.

Computational simulation indicated that TNS exhibited promising alignment with the template 14-α demethylase bound with lanosterol (PDB: 4LXJ). Although TNS lacks the 14-α-methyl group, it exhibited an excellent binding affinity to the 14-α-demethylase enzyme, which is considered to be the key difference with lanosterol. TNS showed a comparable binding energy via multiple hydrophobic interactions with the binding site. Therefore, TNS can be considered as a new class of antifungal that may overcome resistance mechanisms.

Additional future plans include the fraction of other plant extracts, including dichloromethane and ethyl acetate extracts, to identify other phytochemicals. Furthermore, activities other than those reported previously for compounds **2**, **3**, and **4** will be investigated.

## 4. Conclusions

*T. apollinea* is traditionally known for several biological activities, including its antifungal activity. The activity of the plant extract was mainly attributed to prenylated flavonoids. However, in this study, we have reported for the first time a novel steroid derivative, TNS, purified from the plant leaf extract and with a more superior anti-*Candida* activity. In silico simulation, including pharmacophore modeling and docking study, suggested that TNS might be a new *Candida* 14-α-demethylase enzyme inhibitor. The compound has never been reported in nature; however, a much-related compound was identified once from *Aspergillus* sp. fungus, isolated from gorgonia. The compounds, TNS, and the one isolated from *Aspergillus* sp. differ in one double bond. The difference in structure may be attributed to the difference in either the enzymatic system utilized by both organisms and/or the environmental condition adapted by both organisms. Further study to investigate the biosynthetic pathway may useful for potential discovery.

## 5. Material and Methods

### 5.1. Preparation of Plant Material

*T. apollinea* (Delile) DC. (Fabaceae) were collected from a mountainous habitat around Sharjah, Kalba road (24.951257, 56.163090), in August 2019. The plant was taxonomically identified by Prof. Ali El-Keblawy, College of Sciences, University of Sharjah. A voucher specimen was deposited at the University of Sharjah Herbarium. Plant leaves were separated and air-dried in the shade at room temperature. The dried plant leaves were then powdered mechanically by an electric blender (Braun Multiquick hand blender and mixer, Melsungen, Germany).

### 5.2. Extraction and Compound Isolation

The dried plant leaves of *T. apollinea* (500 g) were extracted in distilled methanol (3 times X 1L) at room temperature for 5 days, and were then filtered. The crude extract was evaporated in vacuo at 45 °C until dryness to give 10 g of dried total methanol extract. The methanol extract was then fractionated successively in *n*-hexane, dichloromethane, followed by ethyl acetate (EtOAc). The *n*-hexane fraction (3 g) was applied to a silica gel column with hexane/EtOAc mobile phase gradient from 10% to 100% to give five main sub-fractions. From sub-fraction II (350 mg), compound **1** (60 mg) was isolated and purified by crystallization using methanol. Sub-fraction III (230 mg) was purified over preparative TLC using solvent system hexane/EtOAc (9:1) as the mobile phase to separate compound **2** (100 mg). Compound **3** (150 mg) was isolated and purified by crystallization from sub-fraction IV (150 mg). The schematic diagram of the chromatography separation is shown in Figure 12.

### 5.3. Purification and Identification of T. appolinea Compounds

The compounds were separated on silica gel TLC plates (Kiesegel 60 F254, Merck, Kenilworth, NJ, USA). The compounds were detected under UV at 254–365 nm or by spraying with 10% H_2_SO_4_ in methanol, followed by heating at 120 °C. Column chromatography was performed using silica gel 60 (0.2–0.5 mm, 0.04–0.063 mm, Merck). The ^1^H and ^13^C NMR spectra were recorded on Brucker spectrophotometer AMX400, AV400 MHz instruments (Bruker Biospin GmbH, Rheinstetten, Germany) at a frequency of 500 MHz for ^1^HNMR and 125 MHz for ^13^CNMR. Topspin 3.2 (Bruker Biospin GmbH, Germany) was used for the data acquisition. ESI-MS was recorded on a Bruker Daltonics mass spectrometer coupled with a LCQ DECA XP mass spectrometer (Thermo Electron, San Jose, CA, USA) with an electrospray ionization source (ESI). Data analysis was performed with Excalibur ver. 1.3 or 1.4 software (Thermo Electron, San Jose, CA, USA).

### 5.4. In Silico Target Identification of the Newly Discovered Compound

#### 5.4.1. Generation of Common Feature Pharmacophore

The 3D structures of lanosterol, eburicol, obtusifoliol, and TNS were drawn using Biovia Discovery Studio 2.5 sketching tools (BIOVIA, Dassault Systèmes, San Diego, CA, USA). CHARM forcefield/MMFF94 was applied on the drawn structures. The 3D structures were prepared using the ligand preparation protocol. The common pharmacophoric features between the 3D structures of the 14-α-demethylase enzyme substrates, including lanosterol, eburicol, and obtusifoliol, were constructed using common feature pharmacophore generation protocol. The input parameters of the protocol included (i) fast conformation generation, (ii) selected features including hydrogen bond donor, hydrogen bond acceptor, hydrophobic feature, (iii) maximum pharmacophore generation 10, (iv) minimum inter-feature distance 2.97, and (v) maximum exclude volume 0.

#### 5.4.2. Mapping of the TNS Compound-Pharmacophore Feature

Mapping the 3D structure of TNS with the generated pharmacophores was conducted using ligand pharmacophore mapping protocol in Biovia Discovery Studio 2.5. The input parameters of the protocol included (i) the best mapping only true, (ii) maximum omitted feature 1 and fitting method, and (iii) flexible. The resulting alignment was described by the fit value. Fit value is the measure of how well the ligand fits the pharmacophore. The higher the fit score, the better the match.

### 5.5. Homology Modeling and Molecular Docking Study of the TNS Compound against Candida 14-α-Demethylase Enzyme

Discovery studio 2.5 (BIOVIA, Dassault Systèmes, San Diego, CA, USA) was employed in this study. The sequence of *C. auris* 14-α-demethylase was downloaded as Fasta file (NCBI Reference Sequence: XP_028891800.1). The X-ray crystal structure of *Saccharomyces cerevisiae* 14-α demethylase bound with lanosterol (PDB: 4LXJ) was employed as a template [25]. Align sequence protocol in Discovery Studio 2.5 was conducted using fast pair wise alignment, Blosum scoring matrix, gap opening penalty 10, and gap extension penalty 0.05. The alignment showed 83.4% sequence similarity and 64.9% identity. Prepare Protein Tools was applied on *S. cerevisiae* lanosterol-14-α demethylase complex (PDB: 4LXJ), followed by the application of CHARMm-Momany-Rone force field. The sequence was aligned to the template and Build Homology Model protocols were applied using 4LXJ as a template. The co-crystalized heme and lanosterol were copied from the template into the modeled structure. Optimization levels were set as high. The resulting model number 2 out of 10 models was included in our study as it showed the least PDF Total Energy = 3106.2664 and DOPE Score = −62,615.8. The modeled protein was then verified using MODELER [27] and PROFILE 3D protocols in D.S 2.5. RMSD calculation aligned the 14-α demethylase sequence bound with lanosterol (PDB: 4LXJ) and a homology model equal to 0.942 all protein, 0.57 main chain, 0.543 c-alpha, and side chain 1.25. Visualization of the 3D protein and lanosterol 2D interaction diagram was conducted via Molecular Operating Environment Software (MOE 2020.09, Chemical Computing Group CCG, Montréal, QC, Canada). Docking of the isolated compound and lanosterol within the 3D model of the *Candida* 14-α-demethylase enzyme was conducted using discovery studio 2.5 CDOCKER protocol. The binding site was selected as sphere 10.7179 Å around the substrate heme and lanosterol. CHARMm force field was applied using the ligand partial charge method Momany-Rome [48]. CDOCKER interaction energy was used to evaluate the binding pose. The binding energy calculation was conducted using score and analysis protocol “Calculate binding Energy”.

### 5.6. Anti-Candida Activity of the TNS Compound

The *Candida* strains used in this study included *C. albicans* (SC5314) and *C. auris* clinical isolate (obtained from Centers for Disease Control and Prevention (CDC), Atlanta, South Asian clade, bronchoalveolar lavage). The anti-*Candida* activity of the TNS compound was measured on agar plates and liquid broth media according to modified Clinical and Laboratory Standards Institute (CLSI) and as described by Soliman et al., 2017 [32]. Briefly, 0.1 mL culture containing 10^4^ CFU /mL was streaked on sabouraud dextrose agar (SDA) plates. The plates were then incubated at 37 °C for 24h with filter discs (8 mm diameter) saturated with different compound dilutions (0–1000 μg/mL). The inhibition zones (mm) were measured by determining the diameter of the clear area. Similarly, the activity in the liquid media was measured by incubating the aforementioned concentrations of the compounds into YPD broth media inoculated with 10^4^ CFU/mL in 96-well microplates at 37 °C for 48h. The turbidity representing the microbial growth was measured with a microplate reader (LT-4500, Labtech, Sorisole, Italy) at OD_600_. Each test was performed in triplicate. Fluconazole (Cat# F8929, Sigma-Aldrich, Burlington, MA, USA) was employed as a positive control. YPD cultures containing the vehicle (dimethyl sulfoxide, DMSO at 0.1%) without compounds or antimicrobials were employed as the negative controls. The % inhibition activity of the compounds was calculated relative to the negative control.

### 5.7. Cell Toxicity Assay Using MTT Staining

The cell viability was performed using a 3-(4, 5-dimethyl thiazolyl-2)-2,5-diphenyltetrazolium bromide (MTT) assay, as described before [22]. Briefly, 96-well micro-plates were seeded with normal human fibroblast cell line (HDF, 106-05A, Sigma-Aldrich) (4000 cells per well) for 24 h. The TNS compound at 0–1000 µg/mL was suspended in DMEM media supplemented with 10% FBS and 1.5% penicillin−streptomycin and was filtered and sterilized prior to application on seeded cells. The color change was measured using a Multiskan Go machine (Spectrophotometer) at 570 nm. Each experiment was repeated six times.

Cell viability was calculated using the following formula:

% of living cells = (OD experimental)/(OD control) × 100, while % of cell death was calculated by subtracting the living cells from the total number of cells [49].

### 5.8. Statistical Analysis

The data were collected and graphed using Graph Pad Prism (5.04, La Jolla, CA, USA). The data were analyzed by two-way analysis of variance (ANOVA) using Bonferroni’s Multiple Comparison Test. A *p*-value < 0.05 was considered as significant.

## Figures and Tables

**Figure 1 plants-11-02120-f001:**
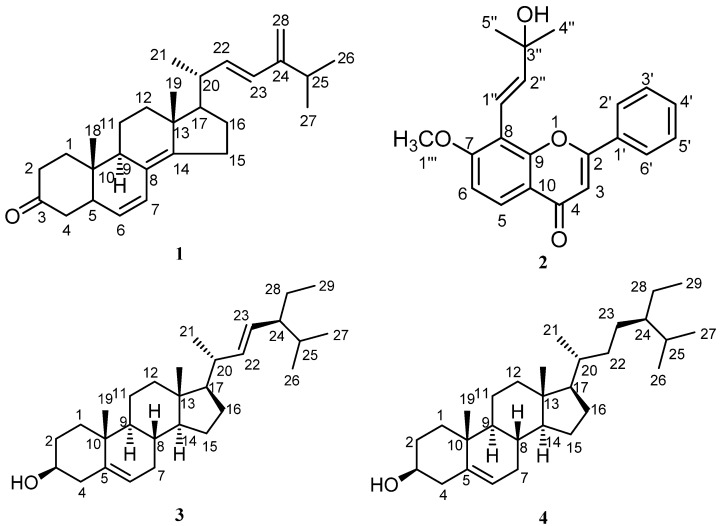
Compounds **1**–**4** isolated from *T. apollinea* leaf methanol extract. Compound **1**; ergosta-6, 8 (14), 22-tetraen-3one, compound **2**; lanceolatin A, compound **3** and **4**; a mixture of stigmasterol and *β*-sitosterol.

**Figure 2 plants-11-02120-f002:**
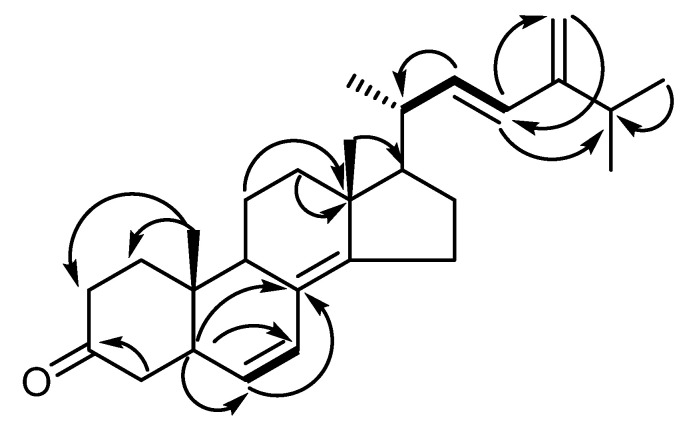
^1^H, ^1^H-COSY (—) and key HMBC (H→C) correlations of compound **1** (TNS).

**Figure 3 plants-11-02120-f003:**
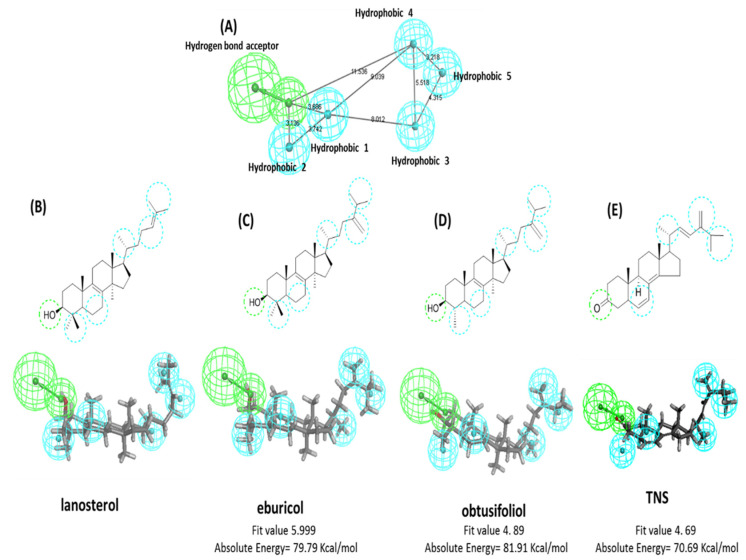
Common features pharmacophore of 14-α-demethylase enzyme substrates and TNS. (**A**) 3D pharmacophore generated from 14-α-demethylase enzyme substrates lanosterol, eburicol, and obtusifoliol. Cyano spheres represent hydrophobic features. Green spheres represent hydrogen bond acceptor features. Lines represent distance in Angstrom. (**B**) 2D and 3D diagram showing lanosterol alignment and feature mapping to generate a pharmacophore. (**C**) 2D and 3D diagram showing eburicol alignment and feature mapping to generate a pharmacophore. (**D**) 2D and 3D diagram showing obtusifoliol alignment and feature mapping to generate a pharmacophore. (**E**) 2D and 3D diagram showing TNS alignment and feature mapping to generate a pharmacophore.

**Figure 4 plants-11-02120-f004:**
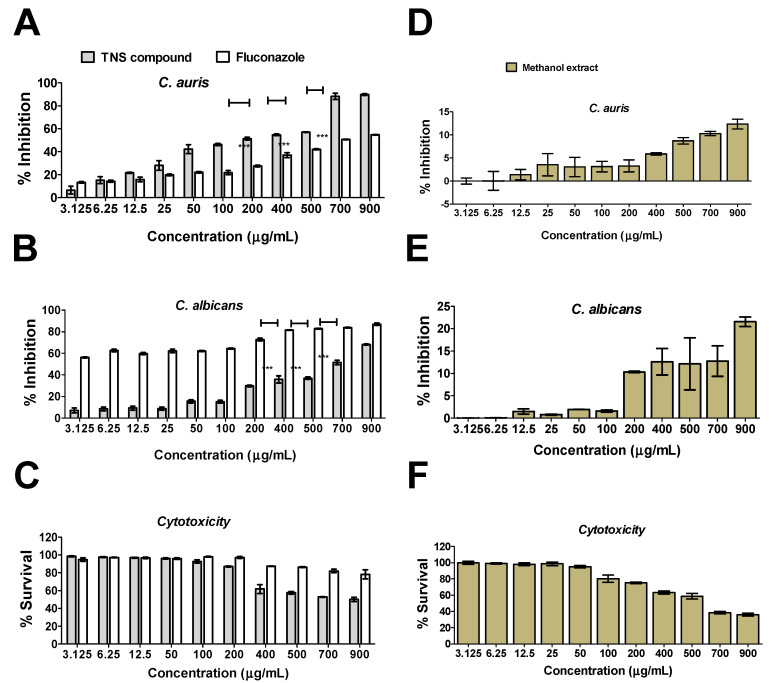
Anti-*Candida* and cytotoxic activities of the TNS compound vs. plant leaves methanol extract. (**A**) *C. auris* growth inhibition by compound TNS at different concentrations. (**B**) *C. albicans* growth inhibition by compound TNS at different concentrations. *Candida* at 10^4^ CFU/mL was grown in 96-well microplates containing YPD broth. The plates were incubated at 37 °C for 24 h prior to measurement at 600 nm using a plate reader. Fluconazole and cultures containing the DMSO vehicle without the compounds were employed as positive and negative controls, respectively. The % inhibition activity of the compounds was calculated relative to the negative control, while the negative control represents 100% growth. (**C**) Cytotoxic activity of TNS against HDF mammalian cells. TNS was tested on HDF cells using an MTT assay. Mammalian cells were seeded in a 96-well plate until confluency, followed by incubation with the samples overnight prior to the MTT assay. (**D**,**E**) Growth inhibition of *C. auris* and *C. albicans* by plant leaf methanol extract at different concentrations. (**F**) Cytotoxic activities of methanol extract on HDF cells using an MTT assay. The data display the mean percentage of the survival rate of mammalian cells ± SEM of five replicates. The data were analyzed using two-way ANOVA and the statistical significance between means was calculated with Bonferroni’s multiple comparisons test. The significance levels are indicated by asterisks. A *p*-value < 0.05 was considered as significant.

**Figure 5 plants-11-02120-f005:**
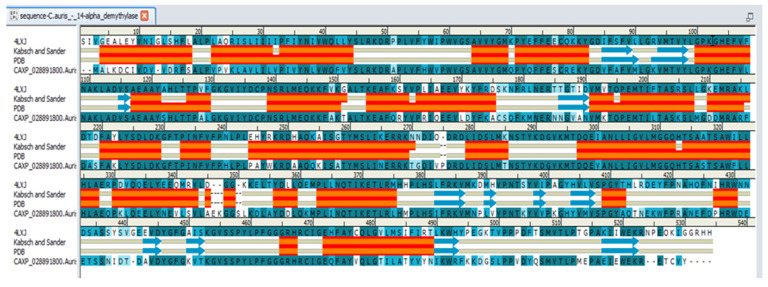
Sequence alignment between *C. auris* 14-α-demethylase (NCBI Reference Sequence: XP_028891800.1) and *S. cerevisiae* (PDB: 4LXJ) using align sequence protocol in Discovery Studio 2.5. Amino acid identity is shown in dark blue, and similarity is represented in light blue. Secondary structure of *S. cerevisiae* 14-α-demethylase is shown. Orange and red zones represent α-helix, blue arrows represent β-sheet.

**Figure 6 plants-11-02120-f006:**
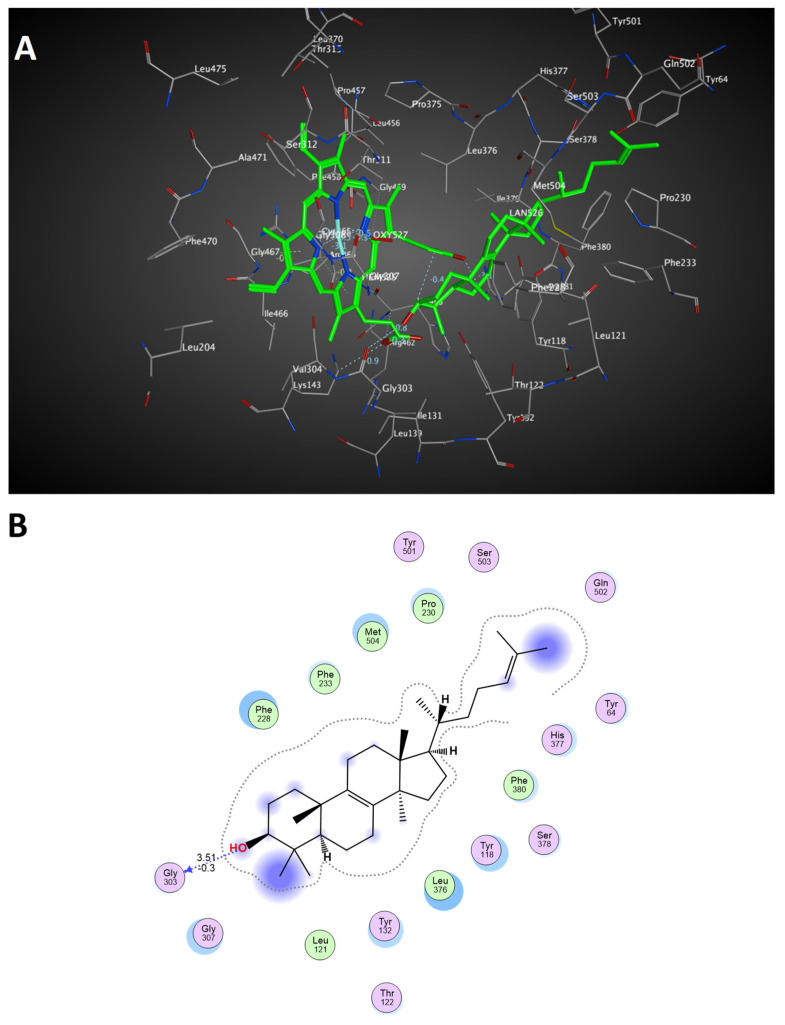
Predicted 3D binding site of *C. auris* 14-α-demethylase enzyme bound with heme and lanosterol: (**A**) 3D representation of binding site including heme and the natural substrate lanosterol (colored in green); (**B**) 2D interaction diagram of lanosterol with the binding site of *C. auris* 14-α-demethylase enzyme. Amino acids in contact with lanosterol are displayed as spheres. Violet spheres represent polar amino acids, while green spheres represent hydrophobic amino acids. Solvent-exposed regions of lanosterol are highlighted in blue. The hydroxyl group of lanosterol showing weak hydrogen bond interaction with glycine 303 (−0.3 Kcal/mol, distance 3.51 Å.

**Figure 7 plants-11-02120-f007:**
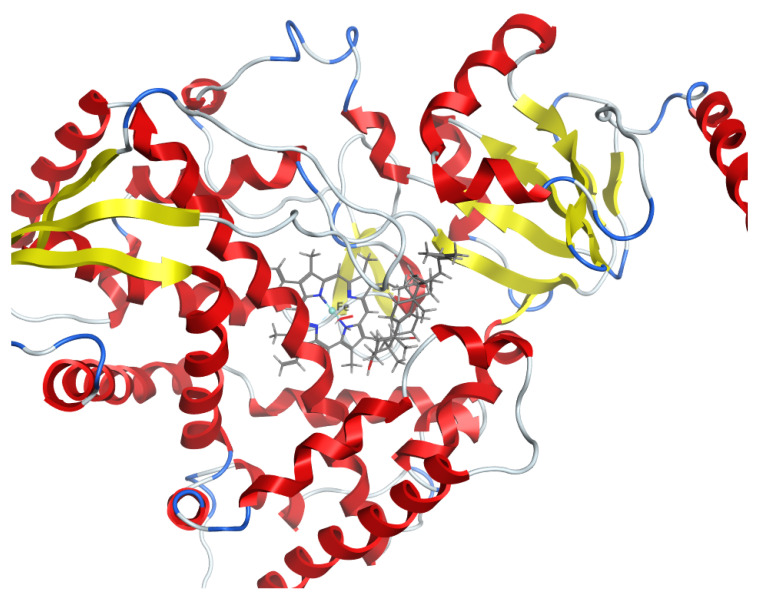
Three-dimensional structure of *C. auris* 14-α-demethylase enzyme bound with heme and lanosterol.

**Figure 8 plants-11-02120-f008:**
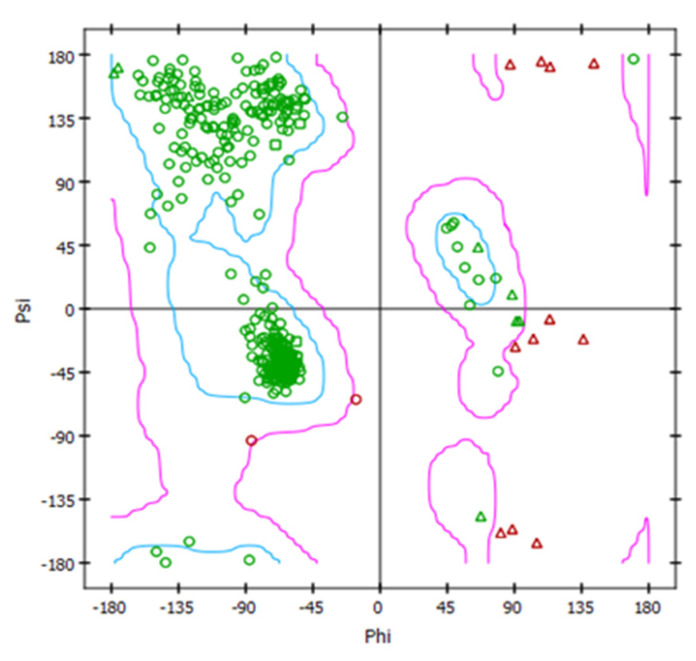
Ramachandran plot of the modeled *C. auris* 14-α-demethylase enzyme.

**Figure 9 plants-11-02120-f009:**
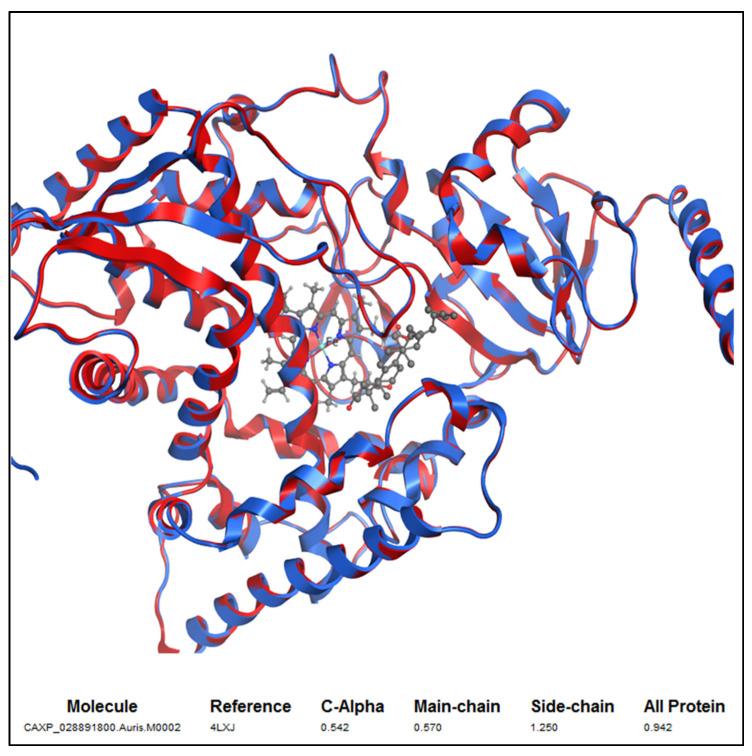
Alignment of the 3D model *C. aurus* colored in red with X-ray crystal structure of *S. cerevisiae* 14-α demethylase bound with lanosterol PDB: 4LXJ colored in blue. Total protein sequence alignment RMSD equal 0.942 using 4LXJ as a reference.

**Figure 10 plants-11-02120-f010:**
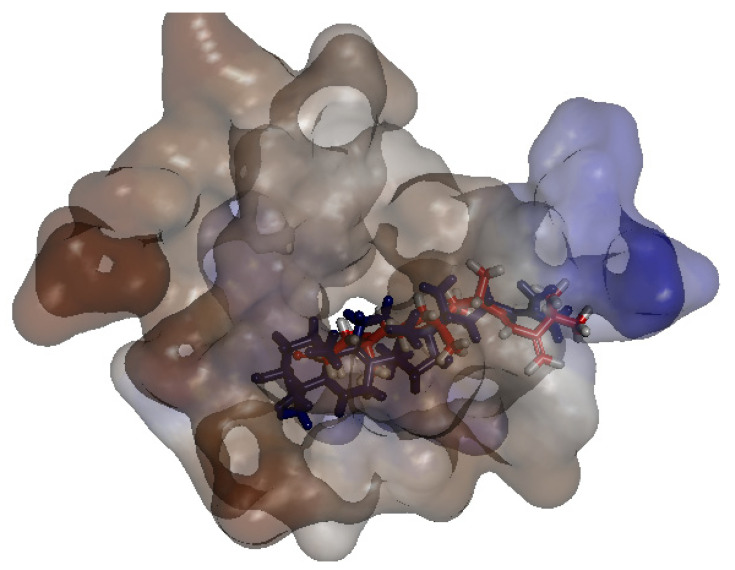
Binding mode of TNS (colored in red) and lanosterol (colored in blue) docked with the *C. auris* 14-α-demethylase enzyme. The binding site is represented as the surface colored by hydrophobic properties. Blue regions represent hydrophilic residues, while the brown color represents hydrophobic residues.

**Figure 11 plants-11-02120-f011:**
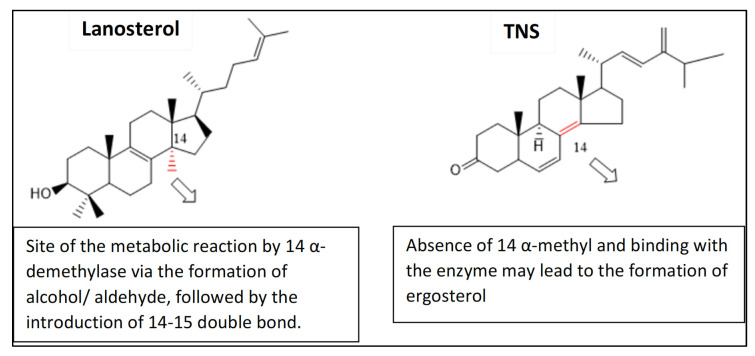
The proposed effect of the structural difference between TNS and lanosterol on ergosterol biosynthesis.

**Figure 12 plants-11-02120-f012:**
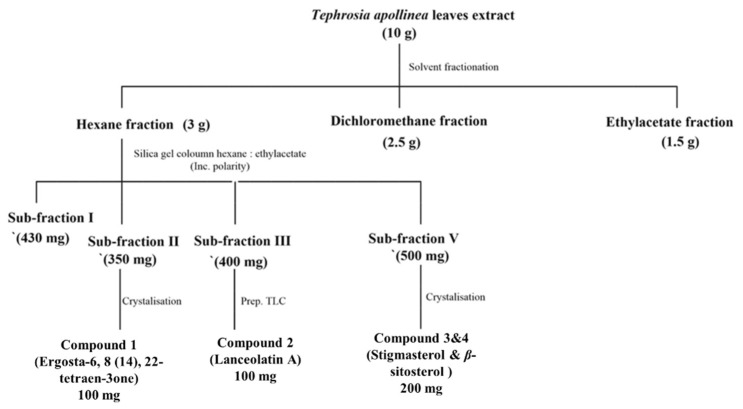
Schematic diagram of the phytochemical isolation of compounds 1–4.

**Table 1 plants-11-02120-t001:** ^1^H- and ^13^C-NMR Data (500 and 125 MHz, respectively; in CDCl_3_) for TNS. *δ* in ppm, *J* in Hz.

Position	*δ* (H)	*δ* (C)	Position	*δ* (H)	*δ* (C)
1	2.04–1.99 (*m*),1.64–1.62 (*m*)	34.7	16	1.69–1.67 (*m*), 1.24–1.20 (*m*)	26.9
2	2.40–2.36 (*m*)	34.6	17	1.59–1.53	32.5
3	--------	207.2	18	0.86 (*s*)	19.4
4	2.21–2.10 (*m*)	40.8	19	1.11 (*s*)	14.2
5	2.38–2.04 (*m*)	45.5	20	1.99–1.97 (*m*)	39.8
6	5.25 (*dd*, *J* = 15.8, 9.9)	129.8	21	0.89 (*d*, *J* = 6.8)	20.8
7	5.13 (*d*, *J* = 9.9)	133.6	22	5.11 (*dd*, *J* = 15.7, 8)	124.4
8	--------	149.1	23	5.78 (*d*, *J* = 15.7)	135.6
9	--------	45.5	24	--------	135.2
10	--------	37.23	25	1.99–2.03 (*m*)	53.0
11	1.69–1.67 (*m*), 1.58–1.55 (*m*)	22.8	26	0.81 (*d*, *J* = 6.8)	19.5
12	1.61–1.59 (*m*), 1.44–1.40 (*m*)	32.1	27	0.81 (*d*, *J* = 6.8)	19.4
13	--------	39.8	28	4.79, 4.74 (*d*, *J* = 2.3)	109.1
14	--------	148.9			
15	2.13–2.10 (*m*), 2.08–2.05 (*m*)	26.6			

**Table 2 plants-11-02120-t002:** Binding energy calculation of TNS-*C. auris* 14-α-demethylase enzyme complex vs. lanosterol−*C. auris* 14-α-demethylase enzyme complex.

Ligand	Binding Energy	Total Binding Energy	Ligand Energy	Protein Energy	Complex Energy	Entropic Energy	Ligand Conformational Energy	Ligand Conformational Entropy
TNS	−49.22	−45.13	87.32	41,302.00	41,339.74	20.20	4.09	0.62
Lanosterol	−46.65	−45.82	132.76	41,302.00	41,387.74	20.40	0.83	0.83

Energy unit is kcal/mol. Results are average of top 10 poses retrieved from docking protocol.

## Data Availability

All data are made available.

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
