# Peer review of "The Anti-Candida Activity of Tephrosia apollinea Is More Superiorly Attributed to a Novel Steroidal Compound with Selective Targeting"

_plants, 2022, doi:10.3390/plants11162120_

Round 1
Reviewer 1 Report
The article "The anti-Candida activity of Tephrosia apollinea is more superiorly attributed to a novel steroidal compound with selective targeting" is an interesting paper. Remarks and suggestions to the authors are:
Title on page 3: 2.1. Change "Discovery of novel phytochemical from T. apollinea that is never reported in plants" to Novel phytochemical from T. apollinea or similar and write in the text that it is a new phytochemical that has not been discovered so far.
In the title Pictures 1, write the names of compounds 1-4.
Title: "2.2. The traditional anti-Candida activity of the plant is more superior owing to the novel TNS compound", please change the title. According to the title, the reader could conclude that the compound was not present in the plant before, but it was, it only has not been discovered yet.
Title: "2.2.1. In Silico target analysis indicated a similarity between TNS and Candida 14-α- demethylase enzyme substrates" please change it. I don't think the title can be the conclusion of the section. It is the same as title 2.2.2. where the section with such a title would compare with previous reports. Same with other titles, please check and change.
The sentence on page 7: "Collectively these results indicate that the traditional anti-Candida activity of T. apollinea is more superior due to TNS" is incomprehensible because TNS was already present in the plant, just before it was discovered. Also, a comparison with previous reports should then be made in the section with such a title.
The sentence on page 7: "TNS at its MIC50 showed no toxicity to mammalian cells (Figure 4C)." Please specify which cells are involved. Do the same in the image and image title.
In Figure 4, in the graphs, please add experiments on untreated cells (if done) and necessarily the influence of the solvent on the cells in the concentration in which it was present at the highest concentration of TNS, i.e. methanol extract and fluconazole.
Title of Figure 4: transfer part of the text to the experimental part if it is not listed there.
Please, in the discussion, show the data (calculate) how much of the pure compound TNS is in the methanol extract at the MIC50 of the methanol extract and then compare the activity.
Section 5.2. Please indicate the temperature and time of leaf extraction with methanol.
Figure 12. Where is compound 4 on the graph? Please add the names of the compounds with the numbers.
Section 5.6. Please specify the solvent in which the compounds were dissolved for testing anti-candida activity.
Please explain in the Discussion:
Why was the biological activity of compounds 2,3 and 4 not investigated? Has it already been described in the literature?
Why was the biological activity of the methanol extract and not the hexane fraction tested? Did you identify the compounds in the dichloromethane fraction and the ethylacetate fraction?
Author Response
Comments and Suggestions for Authors
The article "The anti-Candida activity of Tephrosia apollinea is more superiorly attributed to a novel steroidal compound with selective targeting" is an interesting paper. Remarks and suggestions to the authors are:
Comment#1: Title on page 3: 2.1. Change "Discovery of novel phytochemical from T. apollinea that is never reported in plants" to Novel phytochemical from T. apollinea or similar and write in the text that it is a new phytochemical that has not been discovered so far.
Response: It has been modified as suggested. Thank you
Comment #2: In the title Pictures 1, write the names of compounds 1-4.
Response: The Name of the compounds have been added as suggested.
Comment #3: Title: "2.2. The traditional anti-Candida activity of the plant is more superior owing to the novel TNS compound", please change the title. According to the title, the reader could conclude that the compound was not present in the plant before, but it was, it only has not been discovered yet.
Response: The title has been changed to “TNS compound showed promising anti-Candida activity”.
Comment #4: Title: "2.2.1. In Silico target analysis indicated a similarity between TNS and Candida 14-α- demethylase enzyme substrates" please change it. I don't think the title can be the conclusion of the section. It is the same as title 2.2.2. where the section with such a title would compare with previous reports. Same with other titles, please check and change.
Response: Both titles have been changed to “In silico target analysis of TNS compound” and “Anti-Candida activity of TNS compound”
Comment #5: The sentence on page 7: "Collectively these results indicate that the traditional anti-Candida activity of T. apollinea is more superior due to TNS" is incomprehensible because TNS was already present in the plant, just before it was discovered. Also, a comparison with previous reports should then be made in the section with such a title.
Response: The sentence has been changed to make it more comprehensive, while the comparison with previous reports were included in the discussion section.
Comment #6: The sentence on page 7: "TNS at its MIC50 showed no toxicity to mammalian cells (Figure 4C)." Please specify which cells are involved. Do the same in the image and image title.
Response: The name of cells has been add as requested. Thank you
Comment #7: In Figure 4, in the graphs, please add experiments on untreated cells (if done) and necessarily the influence of the solvent on the cells in the concentration in which it was present at the highest concentration of TNS, i.e. methanol extract and fluconazole.
Response: This was clearly identified in the methods section, which made clearer as ” YPD cultures containing the vehicle (dimethyl sulfoxide, DMSO) without the compounds or antimicrobials were employed as negative controls. The % inhibition activity of the compounds was calculated relative to the negative control.”. The negative control represents 100% growth. Further, a similar sentence has been included now in Figure 4 legend.
Comment #8: Title of Figure 4: transfer part of the text to the experimental part if it is not listed there.
Response: More details methods are included in the method section.
Comment #9: Please, in the discussion, show the data (calculate) how much of the pure compound TNS is in the methanol extract at the MIC50 of the methanol extract and then compare the activity.
Response: The following sentence has been included now in the discussion section “TNS also showed ~4 and 23 times less activity against C. albicans and C. auris when compared to the plant methanol extract, while the plant itself contained ~0.2 mg/gm dry weight.”
Comment #10: Section 5.2. Please indicate the temperature and time of leaf extraction with methanol.
Response: The temperature and time were added to the methods section (room temperature and 5 days).
Comment #11: Figure 12. Where is compound 4 on the graph? Please add the names of the compounds with the numbers.
Response: The figure has been modified as suggested.
Comment #12: Section 5.6. Please specify the solvent in which the compounds were dissolved for testing anti-candida activity.
Response: It has been included now as dimethyl sulfoxide, DMSO.
Please explain in the Discussion:
Comment #13: Why was the biological activity of compounds 2,3 and 4 not investigated? Has it already been described in the literature?
Response: Compounds 2, 3 and 4 are reported compounds with well-identified activities, that is why we focused on the newly discovered compound TNS, where it is activity has never been reported.
Comment #14: Why was the biological activity of the methanol extract and not the hexane fraction tested? Did you identify the compounds in the dichloromethane fraction and the ethylacetate fraction?
Response: This is really a good point, we have not fractionated any other solvent extract, but it is a good future plan. Thank you
These sentences have been added as well at the end of the discussion “Additional future plan includes the fraction of other plant extracts including the dichloromethane and ethyl acetate extracts to identify other phytochemicals. Furthermore, activities other than those reported previously for compounds 2, 3 and 4 will be investigated.”
Reviewer 2 Report
The manuscript entitled "The anti-Candida activity of Tephrosia apollinea is more superiorly attributed to a novel steroidal compound with selective targeting" presents studies on new compounds extracted from the leaves of Tephrosia apollinea. Given the extensive experiments that have been performed to characterise the compounds and the novelty of compounds not described by other authors, I highly recommend the manuscript for publication in Plants. I believe it definitely expands the knowledge of biologically active compounds extracted from Tephrosia. Prior to publication, I suggest to change the titles of Results subsections, using a description of methods rather than the statements that are conclusions.
Author Response
Comments and Suggestions for Authors
The manuscript entitled "The anti-Candida activity of Tephrosia apollinea is more superiorly attributed to a novel steroidal compound with selective targeting" presents studies on new compounds extracted from the leaves of Tephrosia apollinea. Given the extensive experiments that have been performed to characterise the compounds and the novelty of compounds not described by other authors, I highly recommend the manuscript for publication in Plants. I believe it definitely expands the knowledge of biologically active compounds extracted from Tephrosia. Prior to publication, I suggest to change the titles of Results subsections, using a description of methods rather than the statements that are conclusions.
Response: We have modified all the results subsections titles as suggested. Thank you
Round 2
Reviewer 1 Report
Dear Editors and Authors,
the work is interesting and has interesting results. I could recommend it for publication, but the authors did not give a clear answer to some of my questions. I don't understand the answer to comment 9, which was the most important one to me because of the possible synergistic effect. It is not clear why the biological effect of the methanol extract was investigated and only the hexane layer was fractionated. The permitted concentration of DMSO in biological research is 0.2 per cent. Thanks to the authors for all the other answers.
Author Response
First, I really appreciate the reviewer's thorough revision.
Second, to address the reviewer's kind comment
- The lower activity of the methanol extract when compared to the purified compound may because indifferent effect rather than synergistic effect by other components
- We have used only hexane extract for further analysis because the amount was reasonable wen compared to other extracts. also the TLC of all extracts indicated the presence of major clear compounds in the hexane extract. Further, a future plan was set to fractionate the other extracts, which is in the process. Moreover, we have tested the activity on methanol extract since it has been reported previously in other reports; therefore, we want to compare the effect of the purified compound to the methanol extract using our own condition.
- Regrading the DMSO concentration, we have used 0.1% as recommended by CLSI.
Thank you